# SVMNET: NON-PARAMETRIC IMAGE CLASSIFICATION BASED ON CONVOLUTIONAL SVM ENSEMBLES FOR SMALL TRAINING SETS

## ABSTRACT

Deep convolutional neural networks (DCNNs) have demonstrated superior power in their ability to classify image data. However, one of the downsides of DCNNs for supervised learning of image data is that their training normally requires large sets of labeled "ground truth" images. Since in many real-world problems large sets of pre-labeled images are not always available, DCNNs might not perform in an optimal manner in all real-world cases. Here we propose SVMnet – a method based on a layered structure of Support Vector Machine (SVM) ensembles for non-parametric image classification. By utilizing the quick learning of SVMs compared to neural networks, the proposed method can reach higher accuracy than DCNNs when the training set is small. Experimental results show that while "conventional" DCNN architectures such as ResNet-50 outperform SVMnet when the size of the training set is large, SVMnet provides a much higher accuracy when the number of "ground truth" training samples is small.

## 1 INTRODUCTION

Deep convolutional neural networks (DCNNs) are powerful tools for multiple tasks of automatic image analysis, demonstrating paramount success and consequently gaining substantial popularity over the past decade. By analyzing the pixels directly, CNNs can be applied to various types of image content without the need to develop task-specific algorithms, and can easily be applied to a broad range of domains with excellent performance (Khan et al., 2020).

One of the major weaknesses of modern DCNNs is their dependence on a large set of examples for training. Cutting-edge DCNNs can have hundreds of layers, each with thousands of trainable parameters. For instance, the common ResNet-50 (He et al., 2016) contains over $2 \cdot 10^6$ artificial neurons. Therefore, to achieve meaningful performance and avoid underfitting, DCNNs normally rely on relatively large training sets.

Training DCNNs normally requires large datasets of labeled ground truth images. Commonly used datasets include benchmarks such as the Modified National Institute of Standards and Technology (MNIST) or ImageNet. These benchmark datasets provide tens of thousands of images with high-quality annotations for training deep CNNs and are commonly used for testing their performance. However, in many cases of real-world image classification problems, large datasets of clean, labeled ground truth are not available.

A common solution to increasing the size of the training set is data augmentation, in which different modifications of the images in the original dataset can create more training samples. However, that strategy can also lead to biases by overusing the same examples. In some cases transfer learning can be used to fine-tune neural networks using pre-trained models, but for domains with very small datasets for fine-tuning, the pre-trained models may remain too sensitive to their original task.

In many real-world cases large datasets are not available. For instance, in the biomedical domain the acquisition and annotation of each image requires the use of costly medical instrumentation, technicians, and medical staff who can annotate each sample manually. Additionally, human protection procedures and protocols are required for the acquisition of each sample, making the preparation of large datasets less practical. Therefore, biomedical image datasets are normally far smaller than the

modern datasets commonly used to train DCNNs. Rare cases can also make it difficult to acquire a suitable training set (Zhang et al., 2019). The need for a large number of training samples is a practical downside of DCNNs, making it difficult to use optimally in many real-world cases.

The problem of small training sets has been addressed in the past by using previous knowledge for a few shot training (Zhang et al., 2019) and even one-shot training (Fei-Fei et al., 2006; Wolf et al., 2009; Koch et al., 2015; Vinyals et al., 2016; Bender et al., 2018). This paper explores a new form of image classification in cases when the number of samples is limited. Based on an ensemble composition of support vector machines (SVMs), the method can work with no prior knowledge, in a similar manner to "standard" supervised machine learning. Inspired by CNN architecture, SVMnet utilizes a large number of small SVMs to quickly analyze image patches, structured in layers that allow for stacking or custom ensemble techniques. An SVM (Cortes & Vapnik, 1995) is less sensitive to high-dimensionality feature spaces (Chen et al., 2005; Mukkamala et al., 2002), and has the advantage of learning from a relatively small number of training samples (Shin et al., 2005; Byvatov et al., 2003; Paiva et al., 2018) compared to other supervised machine learning approaches.

## 2 ARCHITECTURE OF SVMNET

The proposed SVMnet[1] architecture is designed as a stacked ensemble of numerous simple SVM classifiers organized into one or more layers. Each layer is an array of SVMs which functions similarly to a convolutional layer in a CNN. Each SVM in a layer is independent and all are assigned an equal-sized patch of the layer's input, referred to as a window. Variable stride length and padding, as described in Chapter 2 of Dumoulin & Visin (2018), are specified as hyperparameters. Each input to the following layer is the output of one SVM.

When a layer is evaluated, each SVM in the layer is trained on ground truth labels. The input to the SVM is the flattened portion of each input image that is within the SVM's window. Each pixel channel within the window is essentially treated as one input feature. For instance, a $5 \times 5$ window would create a 25-feature SVM for grayscale input and a 75-feature SVM for 3-channel RGB input. During this step, the SVMs may be given weights based on the accuracy of the fit, used for ensemble classification. Each SVM then predicts a class label or a vector of class probabilities for its window of each input, creating an input tensor for the next layer.

Figure 1 shows a simple layer in SVMnet. Each node in the layer is one SVM, trained using the ground truth labels for the input samples. The weights are determined based on the classification accuracy of the SVM compared to the ground truth of the training set. The weight function is configurable and will be described later in this Section.

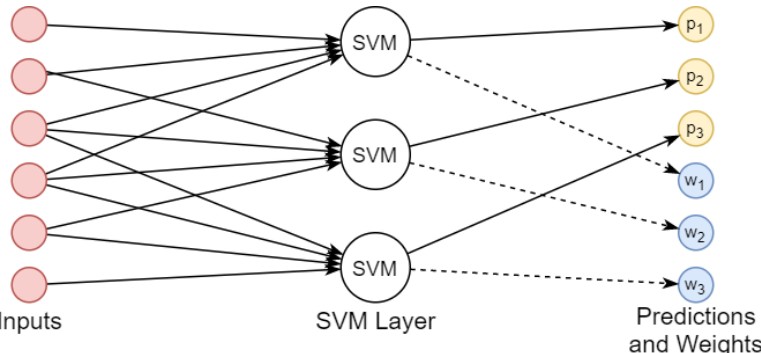

Figure 1: Example of a simple weighted layer of SVMnet. Each node in the layer is an SVM, trained with a subset of the inputs (pixels). Weight outputs are optional for a given layer.

To produce one class label for each input, SVMnet may perform a weighted vote after the final layer. This vote combines the results of the final layer by treating each value as a vote for that class label. If the final layer is weighted, these are used to weigh the votes in favor of SVMs with higher accuracy.

---

[1]SVMnet is not a "network" in the same sense as CNNs and are named as such mostly due to analogy.

$$S_c = \Sigma_i \eta(A_i)[P_i = c] \tag{1}$$

The total voting score $S_c$ of each class $c$ is calculated by Equation 1, where $A_i$ is the accuracy score of SVM $i$ in the final layer, $\eta$ is the weight function, and $P_i$ is the class label predicted by SVM $i$. That is, if the predicted label $P_i$ of SVM $i$ is class $c$, the weighted score $\eta(A_i)$ is added to the vote for that class. The weight function emphasizes the predictions of the SVMs with higher accuracy during training. The class that has the highest score $S_c$ is chosen as the predicted label by the model for the given sample. The weight function $\eta$ is configurable and in our experiments is defined as $\eta(x) = x^2$, where $x$ is the classification accuracy of the SVM determined during training.

While the layers support arbitrary estimators, here we use only support vector machines (SVM), hence the name SVMnet. The SVMs are trained with a Radial Basis Function (RBF) kernel (Chang & Lin, 2011) and scaling gamma value, and they continue to iterate until convergence with a 0.001 tolerance. The ability to choose different estimators in each layer can be compared to the ability to use different activation functions in the layers of neural networks.

Figure 2 illustrates one possible two-layer SVMnet architecture. Each SVM in the first layer analyzes a specific patch of each image and is fitted independently against ground truth labels. These SVMs then produce a vector of class probabilities for the same pixel region which forms the input matrix for the following layer. The SVMs in the second layer are fitted on a region of these probabilities and predict a class label for the image. These labels are then tallied in a final vote to produce one label for the input. The motivation for multiple layers is that secondary layers can in essence learn which of the SVMs in their window are more accurate or "trustworthy", as their predictions are being compared to ground truth labels in each layer.

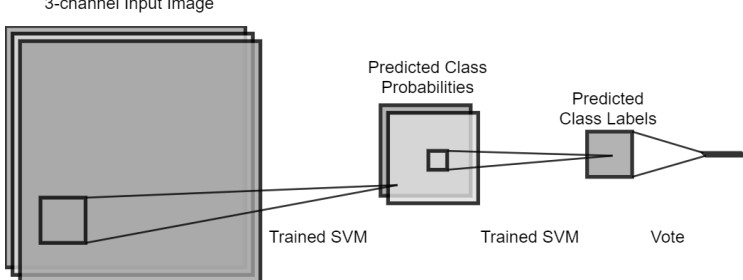

Figure 2: Example SVMnet architecture containing two SVM layers and a class label vote.

## 2.1 DROPOUT

Not every patch is expected to produce a well-informed SVM. Some regions of the images, particularly towards the edge, often lack the details necessary to distinguish samples from each other. This can cause the outputs of these SVMs to act as noise in a vote tally. Even with the expected low accuracy score of the SVM depressing the weight of its vote, if the low-information regions are large then enough inaccurate votes may overwhelm the more accurate votes. To help prevent this, a dropout system is implemented for the vote tally.

When using dropout, which SVMs to drop are calculated when fitting SVMnet. First, the SVMs are ordered from the highest weight to the lowest. Votes are then cumulatively tallied one SVM at a time with the accuracy of the votes measured between each tally. SVMnet then finds the global maximum accuracy of the cumulative tally. This marks the point where including the votes of the less-accurate SVMs lowers the overall accuracy of the tally, so those SVMs are marked for dropout and are not included in the final vote.

## 3 EXPERIMENTAL RESULTS

To test the efficacy of SVMnet compared to a "conventional" CNN, several experiments were performed using common, relatively small datasets. The purpose of SVMnet is not to outperform CNNs

in the general case, but to achieve higher accuracy when the number of labeled training images is limited. Therefore, the experiments were made with different sizes of training sets to compare the classification accuracy as the training set increases.

The performance of the SVMnet was compared to the performance of residual network, or ResNet, models with 18, 34, and 50 layers (He et al., 2016). ResNet is a powerful architecture that was designed to reduce the number of required training samples for deep learning tasks and has demonstrated excellent efficacy in image classification. Each ResNet model was compared when trained from scratch and when fine-tuned using pretrained ImageNet weights. Following the practice in He et al. (2016), the final convolutional layer is followed by a global average pooling layer, then by a single fully-connected layer with softmax activation and as many units as class labels in the respective task. Models were trained using stochastic gradient descent (SGD) optimization with a linearly decaying learning rate (given by $0.999(1 - s/2) + 0.001$ where $s$ is the training step) and Nesterov momentum of $0.9$. The models were trained for a maximum of 200 epochs but were stopped early if the loss on the validation dataset did not improve by at least $0.01$ over 20 epochs. The number of epochs is limited in order to keep the ResNet training times comparable to SVMnet. The resulting accuracy and training time for each model was averaged over 5 repetitions of each experiment.

While the height and width of inputs can be adjusted for ResNet, the architecture always expects 3-channel RGB color images. Grayscale images were modified for use by ResNet by duplicating the pixel values into three equal channels. This approach was used in Section 3.3 and Section 3.4. Before training and classification by ResNet, images were also passed through a preprocessing filter provided by the Keras library to prepare the data for ResNet models. All inputs were normalized by dividing by the mean and subtracting the variance before being used to train SVMnet. For RGB color inputs, the images were normalized per-channel.

All experiments and analysis presented in this section used the same hardware environment. SVMnet was parallelized across 16 cores of Intel Xeon Gold 6130 CPUs, and ResNet models were trained on an nVidia GeForce GTX 2080 GPU.

## 3.1 COIL-100 OBJECT RECOGNITION

Columbia Object Image Library (COIL-100) is a common dataset used for basic object recognition (Nene et al., 1996). It contains RGB color images of 100 different objects, each photographed 72 times at $5 \deg$ increments about the vertical axis. Background details were removed in all images and the objects are centered and enlarged to fill the frame. Some objects contained in this dataset include coffee mugs, small toy cars, and various fruits and vegetables.

The SVMnet in this experiment used one layer with a $25 \times 25$ window (giving each SVM 1875 input features) and a stride length of 7, followed by a weighted vote with dropout. The SVMnet and ResNet models were fitted with 100-500 training images in increments of 100, each controlled to have an equal number of samples from each object. A separate subset of 200 images was used as validation data for ResNet models.

Figure 3 shows the results of this experiment. When fitted on the smallest training set, containing only one example per object, SVMnet correctly predicted labels for over 60% of the remaining images. With the same training set, ResNet-50 showed about the same accuracy and only pretrained ResNet-34 exceeded SVMnet; however, SVMnet was significantly faster to train in all cases.

## 3.2 IMAGENETTE

Imagenette is a fairly small, 10-class subset of the ImageNet dataset (Howard). Several versions of this dataset exist; here we use version 2 of the 160 px dataset with noiseless labels. Many of these images are rectangular with their shortest side scaled to 160 px. In this experiment, we symmetrically zero-pad each image along its shorter axis to make it square, then downscale the images to have the same dimensions of $160 \times 160$ px.

The SVMnet used here contains one layer with a window size of 22 and stride length 7, followed by a weighted vote with no dropout. Imagenette is pre-divided into training and testing subsets containing 9,469 and 3,925 images, respectively. Models were trained using 20, 40, 80, 160, and

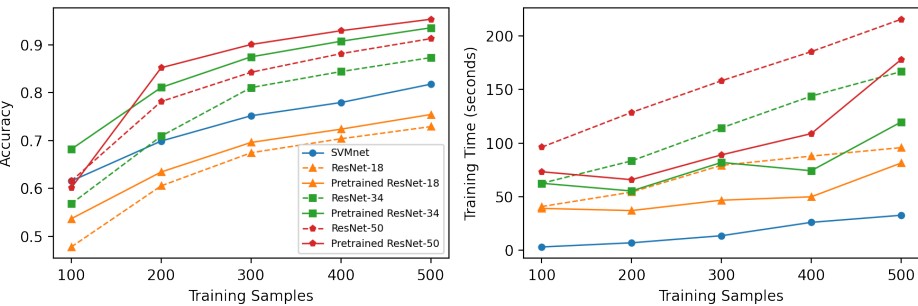

Figure 3: Test-set accuracy (left) and training time (right) of SVMnet and ResNet on COIL-100 images when fitted with different training set sizes.

320 images from the provided training set and evaluated using the provided testing set. An additional 100 images were selected from the training set as validation data for the ResNet models.

Figure 4 shows the results of this experiment. SVMnet achieved higher accuracy than all ResNet models for all training sets except the largest, where the ResNet-50 model pretrained with ImageNet weights improved drastically. The generally low accuracy of these models could be explained by the method used to conform each image to the same dimensions, which introduces a significant amount of empty space in many images. However, even under these conditions, SVMnet attained the highest accuracy in the least time for the smaller training sets.

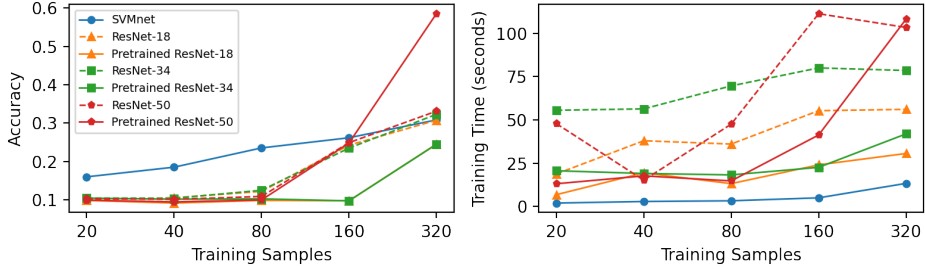

Figure 4: Test-set accuracy (left) and training time (right) of SVMnet and ResNet on Imagenette when fitted with different training set sizes.

## 3.3 COVID-19 RADIOGRAPHY

During the COVID-19 pandemic, machine learning techniques have been applied to various kinds of data to assist the medical community in making accurate diagnoses (El-Din Hemdan et al., 2020; Gangloff et al., 2021; Li et al., 2020). During the early stages of a disease outbreak, diagnostic data is expected to be limited or sparse, making it difficult to train most kinds of machine learning models. A type of model capable of learning from a small number of samples would be the most effective in this time frame.

Here we apply SVMnet to a database of chest x-ray images from healthy patients and patients diagnosed with COVID-19 (Chowdhury et al., 2020; Rahman et al., 2021). In this experiment, only the images labeled as "Normal" and "COVID" are used. Images were downscaled to $128 \times 128$ pixels (approx. 43% of the original size). An equal number of images were selected from each class, totaling 7232 samples. Models were fitted with 10, 20, 50, 100, and 200 training samples, with 50 separate images used as validation data for the ResNet models. The SVMnet uses two layers: the first with window size 19, stride 7, and class probability outputs; the second with window size 5 and stride 5, followed by an unweighted vote. During the architecture experiments described in Section 3.7, the 2-layer SVMnet was shown to outperform the 1-layer models for this dataset.

Figure 5 shows the results of this experiment. SVMnet was able to correctly label between 64% and 78% of unseen x-rays depending on the number of training samples, but most ResNet models failed to make significantly accurate predictions. Only the 18- and 34-layer ResNet models trained from scratch approached the accuracy of SVMnet. Additionally, SVMnet was several times faster to train.

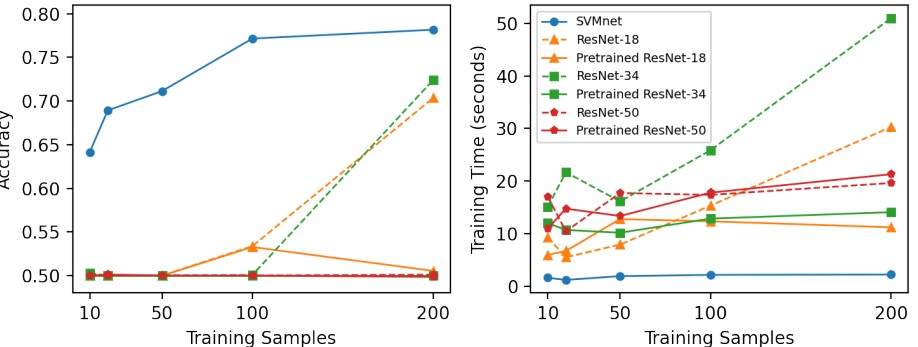

Figure 5: Test-set accuracy (left) and training time (left) of SVMnet and ResNet on COVID-19 chest x-ray images when fitted with different training set sizes. The accuracy of the ResNet models displays considerable overlap.

### 3.4 ASTRONOMICAL IMAGE DATA

To test the performance of SVMnet on a current real-world image classification problem, a dataset of galaxy images from the Panoramic Survey Telescope and Rapid Response System (Pan-STARRS) was used. The dataset is made of galaxies separated into elliptical and spiral morphology. The galaxy images were taken from the catalog of Pan-STARRS galaxies classified by their morphological type (Goddard & Shamir, 2020).

An equal number of images were selected of each morphological type, totaling 26,732 samples. Each image is grayscale and has a dimension of $120 \times 120$ px. SVMnet and ResNet models were fitted with 10, 20, 40, 80, 160, and 320 training samples, with 200 separate images used as validation data for the ResNet models. The SVMnet uses one layer with a window size of 22 and stride 5, followed by a weighted vote with dropout.

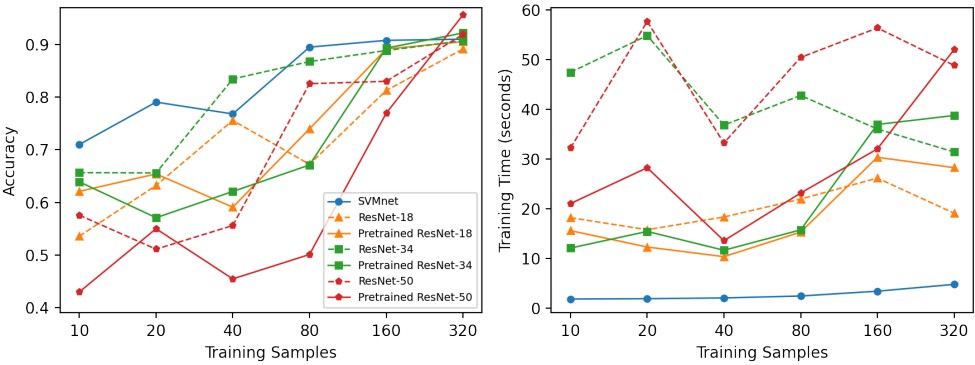

Figure 6: Test-set accuracy (left) and training time (right) of SVMnet and ResNet on Pan-STARRS galaxy images when fitted with different training set sizes.

Figure 6 shows the results of this experiment. As the graph shows, SVMnet outperformed almost every ResNet model when trained with a relatively small dataset. The models generally improve as the training set grows, with several ResNets slightly overtaking SVMnet with the largest training set. In all cases, SVMnet finished training many times faster than all ResNet models.

## 3.5 WND-CHARM

To test a "traditional" approach of using an SVM after extracting image features, we used the WND-CHARM open source feature set (Shamir et al., 2008) combined with an SVM with linear kernel implemented through SVMLib. Table 1 compares the test set accuracy of WND-CHARM and SVM-net using the experimental datasets described earlier in this Section. WND-CHARM was trained on equal sized training subsets and consistently showed lower classification accuracy than SVMnet under the same conditions.

**COIL-100**

|     | WND-CHARM | SVMnet |
| --- | --- | --- |
| 100 | 54% | 62% |
| 200 | 59% | 70% |
| 300 | 61% | 75% |
| 400 | 64% | 78% |

**Imagenette**

|     | WND-CHARM | SVMnet |
| --- | --- | --- |
| 20 | 11% | 16% |
| 40 | 13% | 19% |
| 80 | 16% | 24% |
| 160 | 18% | 26% |
| 320 | 21% | 31% |

**COVID-19**

|     | WND-CHARM | SVMnet |
| --- | --- | --- |
| 10 | 53% | 64% |
| 20 | 55% | 69% |
| 50 | 60% | 71% |
| 100 | 64% | 77% |
| 200 | 66% | 78% |

**Pan-STARRS**

|     | WND-CHARM | SVMnet |
| --- | --- | --- |
| 10 | 52% | 71% |
| 20 | 56% | 79% |
| 40 | 61% | 77% |
| 80 | 63% | 90% |
| 160 | 72% | 91% |
| 320 | 88% | 91% |

Table 1: Comparison of the classification accuracy of WND-CHARM and SVMnet when trained on a small number of samples from four datasets.

## 3.6 COMPUTATIONAL COMPLEXITY

The complexity of fitting an SVM is asymptotic and polynomial. For a training set containing $n$ samples, the algorithm is dominated by either an $n^2$ term or an $n^3$ term based on the formulation of the problem (Bottou & Lin, 2007). Therefore, training a large number of SVMs can be a computationally demanding task, and can lead to substantial computational complexity during training.

The number of SVMs $N$ in a layer receiving rectangular input with width $I_x$ and height $I_y$ is given by Equation 2. The window size $W$ (equivalent to the kernel size in other CNN literature), stride length $S$, and padding amount $P$ in their respective dimensions follow from standard convolutional arithmetic.

$$N = \left( \frac{I_x + 2P_x - W_x}{S_x} + 1 \right) \cdot \left( \frac{I_y + 2P_y - W_y}{S_y} + 1 \right) \tag{2}$$

Fitting a layer in SVMnet requires fitting $N$ SVMs - a polynomial time operation. If the layer includes weights, then the SVMs must predict a class label for each input during the fit step, which scales linearly with the number of samples $n$. When using dropout as described in Section 2.1, SVMnet performs an additional step during training that scales linearly with $n$. Thus, fitting SVMnet is dominated by the polynomial fit time of the SVMs.

CNNs can theoretically be trained infinitely, but there is a definitive point at which the SVMs within SVMnet converge. This places a soft upper bound on the training time of SVMnet based on the tolerance parameter of the SVMs. Additionally, a firm upper bound may be placed on the number of iterations of the SVM algorithm, allowing for a shortened training time at the expense of some accuracy.

SVMnet trains multiple SVMs simultaneously using process-based parallelism and shared memory, greatly increasing its speed on typical multicore computers with minimal overhead. While this

allows SVMnet to run quite easily on relatively inexpensive systems, the potential performance gain from extra hardware is minimal compared to the extreme optimization of CNNs for GPU devices.

### 3.6.1 INFERENCE TIME OF IMAGE CLASSIFICATION

Predicting a single class label of an image using SVMnet typically requires a large number of individual SVMs to predict a label followed by a vote tally. Despite its affinity for parallelization, this process is expected to take longer than the highly optimized matrix operations of a CNN. Table 2 compares the inference time of SVMnet and ResNet on images in the COIL-100 dataset.

|  | 1 | 10 | 100 | 1000 |
|---|---|---|---|---|
| SVMnet | 2.36 | 2.66 | 3.81 | 24.2 |
| ResNet-18 | 0.054 | 0.056 | 0.082 | 0.296 |
| ResNet-34 | 0.060 | 0.060 | 0.098 | 0.410 |
| ResNet-50 | 0.061 | 0.064 | 0.106 | 0.515 |

Table 2: Comparison of the response time (in seconds) of SVMnet and ResNet to predict class labels for 1, 10, 100, and 1000 samples of the COIL-100 dataset.

The comparison shows that SVMnet is significantly slower than ResNet for classifying samples, but the speed of classification is still practical for many real-world systems. The parallelization of SVMnet greatly reduces the time needed to make predictions, but the overhead of shared memory operations is significant in the case of few samples.

### 3.7 ARCHITECTURE COMPARISON

As with CNNs, SVMnet can be configured into a variety of architectures which are expected to differ in performance depending on the classification task. Due to the high number of possible models, determining which is the most effective for a single task is non-trivial. In this section we show how a variety of SVMnet configurations were tested on the COIL-100 dataset to inform the choice of model used in Section 3.1. Similar methods were used to select the models for other datasets. SVMnet models with multiple layers were tested in the same manner.

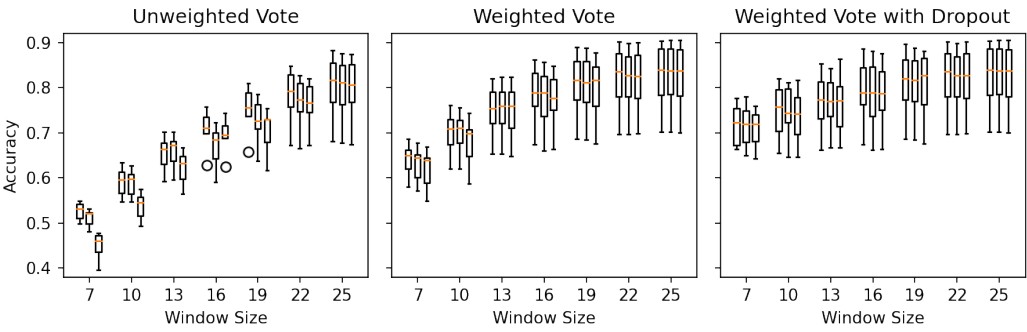

Figure 7: Prediction accuracy of one-layer SVMnet architectures fitted to COIL-100. Each group of three box plots represents the same window size with stride length 3, 5, and 7, respectively. Each box plot shows the distribution in model accuracy when using five training sets of 200-1000 examples.

Figure 7 shows how the performance of a one-layer SVMnet changes with the window size, stride length, voting method, and number of training samples when fitted to COIL-100. Prediction accuracy improves in all cases as the window size increases but with diminishing returns. Increasing the stride length tends to lower accuracy when the window is small but incurs little to no penalty when the window is large. When the vote of an SVM is weighted, model accuracy improves in all cases compared to an unweighted vote; performance increases further when using dropout as described in Section 2.1. This effect is more significant when the window size is small.

# 4 CONCLUSION

Deep convolutional neural networks provide excellent performance in automatic classification of image data while eliminating the need to develop and tailor algorithms for specific image classification problems. With the availability of libraries, DCNNs have become the de facto first solution to image classification.

Here we explore one of the primary weaknesses of DCNNs, which is the need of a relatively high number of labeled "ground truth" samples for effective training of the network. While DCNNs are often tested on relatively large datasets such as MNIST or ImageNet, in many real-world problems a very large number of clean labeled samples that can be used for training is not available.

In many other cases labeled training samples are not available. For instance, when analyzing archaeological artifacts, the number of training samples are limited by the number of available artifacts, which is a hard limit that cannot be easily changed. Using computer vision to analyze art (Khan et al., 2014) is limited by the number of paintings each artists created, which can be a firm limit, especially when the painter is no longer alive.

SVMnet aims at providing an effective solution for the numerous real-world situations in which the number of labeled image samples that can be used for training is limited. SVMnet utilizes the ability of an SVM to learn from a smaller number of samples compared to other machine learning approaches. The flexible structure of SVMnet allows it to learn directly from the pixel values, and to utilize different layers that correspond to the convolutional and fully connected layers in "conventional" deep neural networks. Like DCNNs, SVMnet does not require the design of specific algorithm for a specific image classification problem.

The proposed approach is structured as a network to take advantage of the stronger signal from neighboring pixels, similar to the core idea in the basis of CNNs. SVMs are known for their ability to learn quickly from relatively few training samples. By training many SVMs on small pixel regions across an image, this quick learning can be leveraged to extract much information from small sets of images in less time than it would take to fully train a deep neural network.

Complexity analysis shows that the training time for SVMnet scales more quickly with the number of input samples than DCNNs, suggesting that SVMnet might take substantial computational resources when trained using large datasets. However, SVMnet is designed for situations in which the labeled training set is relatively small. As shown in our experiments, the training time might not be a practical obstacle in many real-world situations in which SVMnet can be used. While computing is an available resource, and training SVMnet with a few hundred training samples scales within reasonable response time, annotated clean or rare training samples might in many cased be much more difficult to obtain.

The underlying structure used to create SVMnet is very flexible, allowing other kinds of machine learning algorithms to be used rather than solely SVMs. Constructing the layers with classifiers such as random forests or logistic regression may result in better performance for some datasets. These layers can be mixed in the same model as well, i.e. using one layer of SVMs followed by a layer of random forests. These possibilities present a promising avenue for future related work.

SVMnet is not designed to become a general solution that can outperform deep convolutional neural networks such as ResNet-50, but experimental results show that it is an effective solution for cases in which the number of labeled training samples is small. Since such cases are not rare, SVMnet can complement "conventional" deep neural networks by providing image classification in the cases where not many labeled training samples are available.

## ACKNOWLEDGEMENTS

The research was funded by NSF grant number XXXXXX.

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
