# OpenReview forum: "SVMnet: Non-parametric image classification based on convolutional SVM ensembles for small training sets"
_ICLR.cc/2022/Conference — ICLR 2022 Submitted_

### Official Review · Reviewer_ZeYm · 2021-10-26

**Correctness:** 3
**Technical Novelty And Significance:** 3
**Empirical Novelty And Significance:** 3
**Recommendation:** 6
**Confidence:** 4

**Main Review:**

In this paper, the author illustrates the following progress:
1. They propose a novel deep learning architecture that uses ensemble of numerous simple SVM classifiers as network layers.
2. They design a voting system for classification along with the SVMNET.
3. In the experiment, they compare the performance of SVMNET with ResNet models with 18, 34, and 50 layers on 3 different datasets. They compare the training time, accuracy versus the number of training samples between these methods and prove the main benefit of the SVMNET.

Questions remain as follows:
1. From the result, generally while the training samples is larger than 200, SVMNET doesn't outperforms the ResNet models except for the
COVID dataset. For most cases, such amount of training data is not so difficult to generate.  So the advantage of SVMNET might be limited.
2. According to the paper, all ResNet models were trained from scratch rather than starting with pre-trained weights derived from ImageNet. In some ways this might be unnecessary since the pre-trained model is available in practice. It would be better if the author can show the comparison result with pre-trained ResNet models.

**Summary Of The Paper:**

One of the weakness of traditional DCNNs is it needs large clean-labeled dataset. In this paper, the author proposes SVMNET, a deep learning architecture which includes a layered structure of Support Vector Machine (SVM) ensembles. The result shows that the SVMNET outperforms other deep convolutional neural networks such as ResNet-50 with less training time for cases in which the number of labeled training samples is small.


**Summary Of The Review:**

In this paper, the author proposes a novel deep learning architecture that uses ensemble of numerous simple SVM classifiers as network layers. Their result shows that the SVMNET outperforms other deep convolutional neural networks such as ResNet-50 with less training time for cases in which the number of labeled training samples is small.
According to the experiment result, it supported the statement to some extent. For two of the three datasets, SVMNET has better performance when the training samples are less than 200 compared with ResNet models, but the ResNet catch up to the SVMNET quickly after the sample increases. Also it would be better if they can do comparison with pre-trained ResNet and show the results in their paper.

---

### Official Review · Reviewer_8kHE · 2021-11-01

**Correctness:** 1
**Technical Novelty And Significance:** 1
**Empirical Novelty And Significance:** 1
**Recommendation:** 3
**Confidence:** 4

**Main Review:**

I will divide my review into three sections, addressing different aspects in the paper: motivation, method and experiments.

Motivation: The paper argues that in order to achieve high performance and avoid overfitting, DCNNs generally require very large labeled training sets. While this claim is true in general, there is a wide array of methods (e.g.: pre-training, self-supervised learning, training on artificially generated samples) previously proposed to address this drawback and I would expect at least some discussion (and experiments) illustrating why the proposed method is important given previous work and how it differs. The paper does mention data augmentation and claims that data augmentation can cause overfitting to overused samples, but this is not true if augmations is done and evaluated correctly (nor is there an experiment demonstrating the claim). The paper also mentioned transfer learning as a solution, but doesn’t discuss it further. Moreover, the paper simply states that the need for large training sets makes DCNNs difficult to use in an optical manner in many real world cases, which is simply not true given the wide applicability of DCNNs in real world applications across numerous fields including medical imaging, autonomous driving, retail, etc. Overall, the motivation for the proposed method and its importance compared to existing work was not well founded.

Method:
In my opinion, there are a few inherent drawbacks to the proposed SVMnet method:
Each SVM corresponds to a specific fixed location in the image. This means that the architecture is sensitive to translation changes or intra class variability (of the location of the object in the image).
The basic building blocks of the architecture are svms which are fed with image patches and instead of being modeled as feature extractors (as in DCNNS), are models as class level classifiers. The result of the first SVM layer are patch level class probabilities, meaning the raw low level visual information of the image is lost (or more precisely encoded only by the output probabilities). Thus, it’s not clear how the svm layers can be stacked in a meaningful way, other than weighing local patch classifiers (and there are no experiments that demonstrate the effectiveness of stacking more than 2 svm layers).

However, it’s worth mentioning that trying to model a network as an ensemble of weak classifiers does make sense, but in my opinion the method has inherent drawbacks and it’s superiority is not demonstrated by the experiments presented.


Experiments:
The authors compare the proposed method against ResNet and claim to achieve competitive performance. However, there are a few major issues in the way the comparison and the experiments were done.

First, the authors state that when training resnet the learning rates for each experiment were set differently and the network was trained for 50 epochs. This value, which dramatically affects the ResNet experiment results, seemed to be set arbitrarily and no method for setting this value (using a validation set used for early stopping, training until convergence, treating the number of epochs as a hyperparameter and optimizing it) is mentioned or explained.

Second, important implementation details for training the ResNet are not mentioned and it seems that no hyperparameter optimization was done, again hindering ResNet performance (which the proposed SVMnet is compared against).

Third, regarding training times - the authors claim to obtain faster running times using SVMNet. However, with having the number of epochs arbitrary set to 50 in ResNet (where SVMNet is not trained in epochs), not mentioning the batch size, checking simply for 50 epochs without considering convergence, early stopping or any meaningful criteria, it’s hard to make sense of the comparison in training times. Also, it’s not clear how the CPU hardware used to train SVMnet compares against the GPU used to train ResNet.

Fourth, the authors compare against ResNet 18, 34 and 50 in settings where the number of training samples is low. However, in such situations, due to risk of overfitting, it makes much more sense to consider architectures with fewer parameters, i.e: fewer layers. Furthermore, in the COIL-100 experiments, it seems that ResNet 18 performs best of the ResNet variants where the performance degrades as the number of layers increases which makes sense since the number of samples is small and ResNets with more layers have more parameters, thus are more prone to overfitting. From this experiment, it seems that a better comparison would be against a ResNet (or any other DCNNs) with fewer parameters (e.g.: fewer layers).

Fifth, COIL 100 is a very constrained dataset, not depicting ‘in the wild’ variability. Specifically, the dataset does not depict large translation variability or large variability in the location of the object in the image, which seems to be the drawback of the proposed architecture. In order to properly evaluate the method, I would recommend using a more challenging benchmark.

Sixth, in the COVID-19 Radiography, all resnet variants receive 50% accuracy (in a binary problem), regardless of the number of samples used for training ,since they predicted the same class for all test images, meaning they are no better than random. This clearly looks like a bug, especially since the very paper cited which presents the benchmark demonstrates almost perfect accuracy for this benchmark with ResNet 18 (given of course the entire training set, but this is evidence that the model does work). The same problem occurs in the Astronomical Image Dataset where when training with 10-80 samples the models achieve random 50% accuracy, but when training with 120 images the performance suddenly increases to 70%-90%.


**Summary Of The Paper:**

The paper argues that one of the major drawbacks for deep convolutional neural networks (DCNNs) is the need for large annotated training sets. To address this drawback, the paper proposes a new architecture, SVMnet, which is designed to achieve relatively high accuracy  (compared to DCNNs) in settings of small training sets.

The SVMnet architecture is composed of one or more stacked "SVM layers". Each SVM layer is composed of a set of independent svm classifiers, where the input to each svm is a patch in the image and the output is a probability estimate for the image class. For example, given a grayscale h X w input image, an SVMnet with 5x5 kernels and stride of 5 would have a 2d “array” of h/5 X w/5 svms, each one trains on a patch of 5x5 pixels whichis flattened to a vector of 25 features. Each svm i is then trained directly to predict the class of the image, based on the 5x5 window, thus resulting in a probability vector for the 5x5 patch. From the validation step, we also obtain the average accuracy of each svm i, denoted by Ai, from which we can compute its weight when aggravating the predictions from all svms. In general each svm layer will have k X l svms, and thh output of the svm layer can be formulated either as a 2d array of k X l predictions or a 3d array of k X l X c of probabilities for each class (where c is the number of classes). When stacking a second SVM layer, the input to that layer is then the k X l X c probability map which is treated as a feature map. Finally, the predictions from the last SVM layer are tallied to produce a majority vote for the image. When training on images with multiple channels, for example RGB images, the channels are flattened to one vector, for example: a 3x5x5 patch would be flattened to a 75 dimensional feature vector.

To demonstrate the applicability of the proposed method, the paper compares SVMnet to ResNet on 3 publicly available datasets and claims to achieve superior performance to resnet in settings of limited training data as well as faster training time.

**Summary Of The Review:**

In my opinion, the experiments presented in the paper fail to demonstrate or give any meaningful evidence to the claims made by the authors regarding the proposed method. While in my opinion the method has inherent drawbacks, the true flaw is the lack of experimental evidence which is a major issue. Also, I believe the claims could have been better supported with a wider discussion demonstrating how the proposed method relates to other previously published work on training deep models in settings with limited training data.

---

> ### Author Response · Authors · 2021-11-22
> **Response to reviewer critique**
>
> Thank you for reading and commenting on the manuscript, and for the insightful suggestions. Numerous changes have been made to the manuscript based on the review.
>
> Based on your suggestions, we have added experiments with pre-trained networks and expanded the discussion on methods developed with few- or one-shot learning and transfer learning. These, however, require previous knowledge, and might not be useful in cases where no previous knowledge exists. Also, unlike methods such as CNN, these methods are not non-parametric, and are not immediate to use. The method proposed here can be applied generically to image datasets in the same way CNNs are used, but more suited for smaller training sets. We do not claim that DCNNs have no real-world applications; clearly, DCNNs have been used and even revolutionized many real-world tasks related to image analysis. But while DCNNs are highly useful, for some tasks the number of training samples is limited, and therefore DCNNs are not fully effective. In many scientific experiments, creating large robust datasets is challenging and expensive, and therefore non-parametric methods that have the advantage when the training set is small can be useful.
>
> In regard to your section about the experimental setup, multiple changes have been made in the revised manuscript. We performed the experiments again, increasing the number of epochs from 50 to 200, adding early stopping criteria and a validation dataset, and clarifying the learning rate. These details were added to section 3. You are correct that COIL-100 is a very constrained dataset. We have added a more challenging benchmark experiment using Imagenette, a 10-class subset of ImageNet, that includes much more variability. The results of this experiment have been added to the manuscript. After having rerun the experiments under updated parameters, the results of the COVID-19 and Pan-STARRS experiments have changed such that we believe your sixth point has been addressed.
>
> Thank you again for your review.

---

> > ### Comment · Reviewer_8kHE · 2021-11-27
> > **Response to Authors**
> >
> > First, I would like to thank the authors for their response and for the revisions made, in particular adding experiments using pre-trained weights and experiments on Imagenette. However, all in all, the results still do not show a significant improvement of the proposed method over existing works, simple baselines or pre-trained ResNets (as shown for example in Figure 3). Adding that to the limited novelty and inherent drawbacks of the method, I do not believe the paper is suitable for publication.

---

### Official Review · Reviewer_SXM2 · 2021-11-02

**Correctness:** 2
**Technical Novelty And Significance:** 1
**Empirical Novelty And Significance:** 1
**Recommendation:** 1
**Confidence:** 5

**Main Review:**

Strenghts:
- the paper presents an ambitious idea by designing SVM classifiers that operate locally. The class probabilities are organized in tensors in a "second layer" which implements ideas from classifier combination (stacked classifiers)

Weaknesses:
- the SVMNet is not strictly a "multi-layer" network since each layer and each local classifier is trained independently. Although the outputs are "connected" in some way there is no joint training or any "chained" learning. Therefore, claiming it is a network it is a bit farfetched.
- the baselines are not good: training from scratch is clearly not the way to go when it comes to ResNets. I don't think the authors can just ignore the existence of pretrained weights. Another baseline would be to train a 2-layer Dense network (with the actual pixels) and a 2-layer CNN network, which would resemble much more the proposal.
- the choice of the datasets is also debatable: why not using MNIST, Fashion-MNIST, ImageNette, or others, which have many results reported in previous papers and would allow for a better comparison?
- training time is emphasized in the first part of the paper, but in fact this is not critical: inference time would be much better as a comparison. In fact, Table 1 confirms this drawback

**Summary Of The Paper:**

The paper describes the use of SVM classifiers trained independently using local receptive fields of images, outputting class probabilities for each pixel that are organized in channels. New independent SVM classifiers are trained over the class probabilities and their results are combined using voting to perform the final prediction. The authors report experiments on two datasets, comparing with versions of ResNet trained from scratch, in which SVMNet shows better results than ResNet when using fewer training examples, For one of the datasets, the ResNet could not converge, while SVMNet obtained up to 80%.

**Summary Of The Review:**

The paper presents an idea of classifier combination much more than an actual "network". The concepts borrowed from multiple-classifier systems and related fields are not novel. Also, the baselines were not adequate, and the choice of datasets to compare was limited. Therefore, I cannot recommend this paper to be accepted.

---

> ### Author Response · Authors · 2021-11-22
> **Response to reviewer critique**
>
> Thank you for reading and commenting on the manuscript, and for the helpful criticism. Numerous changes have been made to the manuscript based on the review. To address each of your points in the order presented:
> * SVMnet is indeed not a network in the same sense as a CNN. The outputs are connected, but each layer is trained independently, and not the entire network as a whole as done in CNNs. The name was chosen based on the analogy to CNN architecture. An explanation has been added to the paper (Section 4), and the language has been clarified in regards to the methods used (stacking and ensemble learning).
> * Thank you for the suggestions. We changed the experiments to include data with both pre-trained and scratch-trained neural networks. We also included a baseline comparing the performance of a "traditional" approach using SVMs on image features.
> * Thank you for the suggestion. An additional experiment was performed on the Imagenette dataset and added to the manuscript.
> * Inference time can indeed be a drawback of the method. However, because the method aims at domains with limited data available, this is not expected to be a practical downside in the intended real-world cases.
>
> Thank you again for your review.

---

> > ### Comment · Reviewer_SXM2 · 2021-11-30
> > **Response**
> >
> > - I agree with the first point, then maybe the more appropriate name would be SVM-Array or something else than a network which can in fact confuse readers.
> > - The experiments improved a bit but still insufficient to show relevant and practical ability of the present method to be scalable beyond what could be achieved with better performance with CNNs.

---

### Official Review · Reviewer_ycpg · 2021-11-02

**Correctness:** 3
**Technical Novelty And Significance:** 2
**Empirical Novelty And Significance:** 2
**Recommendation:** 3
**Confidence:** 4

**Main Review:**

Strengths:
- The problem studied is relevant and importnat to solve. For many applications the amount of labeled data available is small.
- The studied idea is interesting and result are encouraging.
- Sensitivity analysis was performed for the window/patch size parameter.
- Authors tested the approach in multiple datasets.
- The paper is easy to follow.

Weaknesses:
- The evaluation framework is inadequate for the proposed apporach. It is expected for neural networks trained with very small data to quickly overfit. More so the larger the network architecture is, and it is unfair to compare that setup to an SVM ensemble. How is the performance of the neural network when you do finetuning of the last few layers from imagenet pretrained weights, or under stronger regularization? All those are standard approaches used by practitioners.
- Simple baselines like having an svm classifier or an ensemble of SVMs trained on a feature representation of the imagery are missing
- SVMnet's Inference time is much higher compared to standad neural networks.
- There is a large line of work on few shot learning, N shot learning, and even zero shot learning that focuses on this specific problems that are not mentioned in this work. Comparing with those would approaches provide a better evaluation framework.
- The reason to set up multiple layers of of SVM trained only on local patches is not explained beyond emulating neural networks.

**Summary Of The Paper:**

In this work auhors proposed an ensemble of multiple SVM classifiers setup in a hierarchical way (somewhat similar to neural networks). Each SVM is trained is a small patch or window from the input imagery and some classifiers can be eliminated from contributing to the predictions. Results show better performance by the model in the small data regime compared with larger convolutional neural networks trained from scratch.

**Summary Of The Review:**

I think the idea proposed is interesting and tackles an important problem, however the it lacks a proper evaluation setup. I do not think the paper is ready for publication and I recommend the authors to retrink the evaluation framework and add relevant baselines and method for comparison.

---

> ### Author Response · Authors · 2021-11-22
> **Response to reviewer critique**
>
> Thank you for reading and commenting on the manuscript, and for the insightful suggestions. Numerous changes have been made to the manuscript based on the review. To address each of your points in the order presented:
> * We made some major changes to the performance evaluation to address fine-tuning. Namely, new experiments with ImageNet pre-trained networks were performed, and the results have been added to the manuscript.
> * Thank you for the suggestion. We added experiments for comparison with "traditional" approach of an SVM trained using feature representation of the image content. The experiments and explanation of how they were done has been added to the paper.
> * That is correct. In cases were the number of images is small, it is often beneficial for the user to sacrifice response time for higher accuracy. A note has been added to the manuscript (to Section 1 and Section 4).
> * References to previous work, especially one-shot training have been added. The method proposed here does not rely on any prior knowledge, which allows it to be general, similarly to CNN architectures. That has also been added to the paper.
> * That is a good point. A justification for multiple layers was added in Section 2.
>
> Thank you again for your review.

---

### Decision · Program_Chairs · 2022-01-20

**Decision:**

Reject

**Comment:**

The paper proposes to overcome the challenge of annotating datasets to train convolutional networks by considering instead an architecture that is composed of stacked support vector machine layers. Each support vector machine is trained on a small patch from the input image. A voting mechanism is used to aggregate the predictions. Results show better performance by the model in the small data regime compared with larger convolutional neural networks trained from scratch.

The reviewers appreciated the relevance of the problem and the originality of the approach. The reviewers also appreciated several parts of the experimental evaluation that were carefully conducted in particular the sensitivity of the analysis with respect to the patch size and the multiple datasets considered for the experimental evaluation. The reviewers also expressed concerns about the adequacy of the evaluation (unfair comparisons), the completeness of the baselines (missing baselines), and the significance of the improvements. In particular, the experimental evaluation was considered too limited given the problem considered.

The authors submitted responses to the reviewers' comments. After reading the response, updating the reviews, and discussion, the reviewers considered that ‘the experimental evaluation improved a bit', that several concerns were satisfactorily addressed, and yet that the updated results 'do not show a significant improvement of the proposed method over existing works, simple baselines or pre-trained ResNet'. We encourage the paper to pursue their approach further taking into account the reviewers' comments, encouragements, and suggestions. The revision of the paper will generate a stronger submission to a future venue.

Reject.